# A Comparison of Pre- and Post-Clinical Simulation Anxiety Levels of Undergraduate Medical Students Before and During the COVID-19 Pandemic: A Prospective Cohort Study

**DOI:** 10.3390/bs15040447

**Published:** 2025-04-01

**Authors:** Rafael Martín-Sánchez, Ancor Sanz-García, Samantha Diaz-Gonzalez, Miguel Ángel Castro Villamor, Silvia Sáez-Belloso, Joseba Rabanales Sotos, Leyre T. Pinilla-Arribas, Pablo González-Izquierdo, Sara de Santos Sánchez, Francisco Martín-Rodríguez

**Affiliations:** 1Advanced Life Support, Emergency Medical Services (SACYL), 47003 Valladolid, Spain; rafael.martin@uva.es (R.M.-S.); ssaezb@uva.es (S.S.-B.); francisco.martin.rodriguez@uva.es (F.M.-R.); 2Faculty of Nursing, Universidad de Valladolid, 47003 Valladolid, Spain; 3Faculty of Health Sciences, University of Castilla la Mancha, Talavera de la Reina, 45600 Toledo, Spain; ancor.sanz@uclm.es (A.S.-G.); samanta.diaz@alu.uclm.es (S.D.-G.); 4Technological Innovation Applied to Health Research Group (ITAS Group), Faculty of Health Sciences, University of de Castilla-La Mancha, Talavera de la Reina, 45600 Toledo, Spain; 5Evaluación de Cuidados de Salud (ECUSAL), Instituto de Investigación Sanitaria de Castilla-La Mancha (IDISCAM), 45071 Toledo, Spain; 6Faculty of Medicine, University of Valladolid, Edificio de Ciencias de la Salud, Avda. Ramón y Cajal, 7, 47005 Valladolid, Spain; sara.santos@estudiantes.uva.es; 7Faculty of Nursing, University of Castilla la Mancha, 02071 Albacete, Spain; joseba.rabanales@uclm.es; 8Group of Preventive Activities in the University Health Sciences Setting, University of Castilla-La Mancha, 02071 Albacete, Spain; 9Emergency Department, Hospital Clínico Universitario, 47003 Valladolid, Spain; ltpinilla@saludcastillayleon.es (L.T.P.-A.); pgonzaleziz@saludcastillayleon.es (P.G.-I.)

**Keywords:** anxiety, high-fidelity clinical simulation, undergraduate students, COVID-19 pandemic

## Abstract

The aim of the present study was to compare the anxiety of undergraduate medical students who were conducting clinical simulation (CS) prepandemic, during the pandemic, and postvaccination. The participants carried out an emergency simulation in a high-fidelity clinical skills laboratory. A prospective, simulation-based clinical cohort study of sixth-year undergraduate medical students was performed over three time periods: from 1 January to 15 April 2019; from 28 September to 18 December 2020; and from 11 May to 18 May 2022. The primary outcome was anxiety level (pre- and postsimulation) measured with the STAI test. Data on student demographics and baseline vital signs (before CS) were collected. A total of 373 students were ultimately included. A total of 40.2% of the cases were prepandemic (150 cases), 20.4% were pandemic (76 cases), and 39.4% were postvaccination (147 cases). The study period had a statistically significant effect on anxiety. There was a statistically significant increase in the incidence of anxiety during the pandemic time period compared with that during the prepandemic and postvaccination periods; no difference was found between the prepandemic and postvaccination periods. Performing CS in biohazardous environments significantly increases anxiety levels, so establishing mitigating measures to minimize the undesired effects of anxiety and promote the simulation-based learning process is necessary. The study was carried out at a single university; in future studies, it is necessary to carry out multicenter investigations to confirm the results.

## 1. Introduction

The ongoing pandemic caused by coronavirus disease 2019 (COVID-19) has led to a shock to health systems worldwide. The most severely ill patients who developed novel severe acute respiratory syndrome coronavirus 2 (SARS-CoV-2) infection required hospitalization, with a previously unknown rate of intensive care unit admission and highly significantly associated mortality ([13]). This disruption has been carried out in all sectors of society, including unsurprisingly, education, especially advanced education in healthcare sciences ([30]).

On the one hand, during the crisis, in the toughest phases of the pandemic, the quarantine phase, face-to-face education, was not possible ([1]). Even though distance learning (online) is an invaluable aid in studies such as those in the health sciences ([16]), it cannot replace clinical or laboratory practice ([10]) ([34]). With research-derived knowledge about the mode of transmission, pathophysiology, treatment, and mass vaccination ([11]), the incidence of adverse outcomes from this disease is declining, leading to acceptable biosafety conditions for continuing face-to-face education, at least fundamental practices ([3]). On the other hand, certain specialties (e.g., medicine, nursing, midwifery, physiotherapists, and podiatrists) necessarily require face-to-face internships, which are not possible in hospitals for obvious reasons; however, conditions must be provided so that at least students in the final courses can conclude their programs ([29]). In this special scenario, high-fidelity clinical simulation (CS) has become an incalculable resource that can decisively aid in training in the decision-making process at critical moments but can be performed under controlled conditions and with very rigorous biosafety protocols ([9]).

Undergraduate medical students, as part of the final curricular program, must compulsorily rotate in an advanced clinical simulation center. As an educational method, CSs have manifold benefits and are becoming a standard and well-proven teaching strategy in health sciences studies ([25]). In this sense, during a regular simulation training session, undergraduate medical students are faced with the autonomous decision-making process, which includes, among other things, giving orders, managing multidisciplinary teams, evaluating the simulated patient, etc., causing considerable anxiety, just by using the pedagogical method of clinical simulation ([38]; [8]; [17]).

Clinical scenarios (however, simulations) can trigger powerful reactions in healthcare students, generating mixed feelings. As part of this special context, undergraduates must address complex conditions, play an essential role in the decision-making process, and even accept that, despite following all the standard procedures, the simulated patient may not experience a favorable outcome. Clinical simulation provides undergraduate medical students with an opportunity to face this anxiety before they finish regular training in a safe and secure environment ([26]; [4]).

CS is a well-known teaching skill associated with a remarkable load of anxiety and unpredictability. Students must operate way outside of their comfort zone, be observed by classmates and facilitators, have unfamiliar conditions and equipment, and be generally in critical scenarios ([24]).

Therefore, the aim of the present study was to analyze three cohorts taken at different chronological time periods (all undergraduate medical students who were conducting the same simulation scenarios), namely, the prepandemic cluster (2019), pandemic cluster (September to December 2020), and last postvaccination cluster (2022), to test baseline and postsimulation anxiety levels and to determine the impact of COVID-19.

## 2. Materials and Methods

### 2.1. Study Design

A prospective, simulation-based clinical cohort study was carried out with volunteer sixth-year undergraduate medical students at the Advanced Clinical Simulation Center, Faculty of Medicine, Valladolid University (Spain), over the following three chronological time periods: from 1 January to 15 April 2019; from 28 September to 18 December 2020; and from 11 May to 18 May 2022.

The study complied with the Declaration of Helsinki, was approved by the Institutional Review Board of the Public Health Service (reference: PI-033/18), and was registered with the WHO International Clinical Trials Registry Platform (doi.org/10.1186/ISRCTN32132176). This study conforms to the broad EQUATOR guidelines and is aligned with the Strengthening the Reporting of Observational Studies in Epidemiology (STROBE) Initiative Appendix A. Written informed consent was obtained from all the participants. For compulsory clinical simulation practices for final-year medical students, a short prebriefing was held to explain the purpose of the study and its implications. Students who volunteered (and with no compensation in return) were registered and then read and signed the informed consent form.

### 2.2. Biosafety Protocol

In the pandemic cluster, to safeguard the health of everyone, a stringent and meticulous procedure was applied.

The simulation lab extends over 68 m^2^, and the debriefing suite covers 32 m^2^ with a capacity for 16 students. Upon access to the simulation center, an antigen test (nasal swab) was performed via the SARS-CoV-2 Rapid Antigen Test (Roche Diagnostics, Mannheim, Germany), and tympanic temperature and hand washing with antiseptic gel were mandatory for every check-in or check-out at the center; moreover, the use of the FFP2 facemask was mandatory at all times.

In addition, frequent surface cleaning, social distancing (a maximum of 8 students per group and two teachers), forced air ventilation, and air quality monitoring were performed. Finally, in the simulation laboratory, the use of an FFP2 mask, gloves, and face shield was compulsory ([27]).

The normal prepandemic and postvaccination protocols were identical but without biosecurity precautions.

### 2.3. Participants and Randomization

The participants were selected from sixth-year undergraduate medical students interested in collaborating in the study, without receiving any compensation in exchange. Students receiving anticonvulsant or anxiolytic treatment and participants completing the State-Trait Anxiety Inventory (STAI) during the last year were excluded.

The participants were randomly assigned to one of eight simulated clinical scenarios (hip fracture, asthmatic exacerbation, anaphylactic shock, polytraumatized, acute heart failure, sepsis, thromboembolism in pregnant women, and acute myocardial infarction). Nonreplacement randomization with a 1:8 ratio was performed via R, version 4.1.2 (package dply). The participants remained unfamiliar with the scenario up to the time of prebriefing and entry into the simulation lab.

### 2.4. Outcome

The primary outcome was the delta of anxiety level (pre- or postsimulation) on the STAI state and trait subscales according to the timeline cohorts.

The validated Spanish STAI version ([7]) is a self-reported scale composed of two subscales (20 items per subscale), namely, state (how the person feels at that specific moment) and trait (how the person usually feels), and it was used for measuring anxiety levels; these two scales exhibited internal consistency (Cronbach’s = 0.9–0.93) ([14]). Anxiety is understood as an emotional reaction or as a personality feature. Anxiety refers to an individual’s tendency to react anxiously, while anxiety is described as a transitory and fluctuating emotional state over time.

### 2.5. Study Protocol

All staff received specific training in the use of the data collection notebook (using paper questionnaires), the standardized acquisition of vital signs, and the correct completion of the STAI. No masking of the intervention was provided. To minimize cross-contamination, the investigators were unaware of the initial assignment, and only the PI and data manager were familiar with the coding and correction items of the STAI. To guarantee the blinding of the results and the anonymization of the confidential data, only the identification data of the volunteer appeared in the informed consent document. In this document, each volunteer was assigned a four-digit code, which was used successively to identify them. Only the PI had access to the informed consent and the code of each participant.

The participants were randomly assigned to one of the eight simulated clinical scenarios by using a high-fidelity manikin SimMan^®^ ALS (Laerdal Medical AS, Stavanger, Norway).

High-fidelity clinical simulation sessions last approximately ten minutes. Two undergraduate medical students entered the simulation laboratory accompanied by a registered nurse. During the case, one student played the role of the leader and the other played the role of the assistant, performing the case history, clinical examination, and initial treatment of the proposed cases.

A survey containing epidemiological details and baseline restricted-time (STAI) scores was completed on the day of the test. Prerandomization, a registered nurse collected a set of baseline vital signs (oxygen saturation, perfusion index, blood pressure, heart rate, and temperature).

A standardized 20-min presentation, fully videotaped, on the features of the simulation manikin and the correct way to operate in the laboratory, was broadcast before the simulation cases. The simulation was conducted by two-person teams (leader and assistant) along with a registered nurse (with direct radio feedback to facilitators) for a maximum of 10 min. Following every scenario, both participants completed the STAI (unrestricted time), and their final vital signs were collected immediately. Pulse oximetry saturation, systolic and diastolic pressure, heart rate, and temperature were measured via the Connex^®^ Vital Signs Monitor (Welch Allyn, Inc., Skaneateles Falls, NY, USA).

### 2.6. Statistical Analysis

Details regarding the data collection, missing values, and sample size calculations can be found in Appendix A; briefly, sample size calculations were performed via balanced one-way analysis of variance tests. Absolute values and percentages were used for categorical variables, and median interquartile ranges (IQRs) were used for continuous variables. The associations between the study period (prepandemic, pandemic, or postvaccination) were assessed by univariate analysis, ANOVA, the Kruskal–Wallis test, or the chi-square test, when appropriate. To determine the pairwise difference between each of the three groups of time periods, a post hoc analysis (Tukey) was performed for those continuous variables that presented a statistically significant main effect in the univariate analysis. Additionally, a multinomial logistic regression was performed for the study period (including the three groups), and all the variables that presented a *p* value < 0.001 in the univariate analysis were used to determine the variables associated with the study period. All calculations and analyses were performed by using our own codes, R packages, and base functions in R, version 4.2.2 (http://www.R-project.org; the R Foundation for Statistical Computing, Vienna, Austria).

## 3. Results

A total of 373 students fulfilled the inclusion criteria, with no missing data and three dropouts (Figure 1). The median age was 23 years (IQR: 23–24; range: 22–41), and 68.6% of the students were females (256 participants). The population distribution in the respective cohorts was 40.2% prepandemic (150 cases), 20.4% pandemic (76 cases), and 39.4% postvaccination (147 cases). No significant differences were reported for age, sex, leader or assistant, or clinical scenario. Table 1 displays the epidemiological variables and vital signs and the baseline and postsimulation evaluation results. A comparison of baseline characteristics according to the scenario is shown in Appendix A.

### 3.1. Primary Outcome

The STAI state subscale revealed notable differences. The pandemic cluster presented a median of 57 points vs. 41 points in the other clusters; i.e., the students analyzed during the pandemic period presented 28% more anxiety as a baseline. Following the simulation, similar disparities were maintained, with a 32% increase in the level of anxiety in the pandemic cluster vs. the prepandemic and postvaccination clusters (*p* < 0.001 in all cases) (Table 2). The STAI trial results presented fewer differences in the means than did the State trial results but were still significantly different. The level of baseline anxiety as a trait during the pandemic phase was significant, with a median of 41 points vs. 31 or 32 points in the other clusters. This difference was maintained at the end of the simulated scenario (Table 2). Finally, the difference between the postsimulation and baseline data, not surprisingly, yielded the same variation.

### 3.2. Logistic Regression

To evaluate the associations of the parameters analyzed during the study period, logistic regression was performed (Table 3), and the results included previous experience, initial systolic blood pressure, initial mean blood pressure, differences pre- and posttraining for the trait STAI and state STAI, and the study period. Only differences pre- and postintervention for the trait STAI were statistically significant (*p* < 0.001 for both the prepandemic and postvaccination groups). The difference in initial systolic blood pressure was significant only for the prepandemic group (*p* = 0.040).

Figure 2 shows the differences before and after training for the trait STAI and the state STAI. According to the post hoc analysis for the trait STAI, there was a statistically significant increase in the incidence of anxiety between the pandemic and prepandemic (*p* < 0.001) and postvaccination (*p* = 0.001) periods, but no difference was found between the prepandemic and postvaccination periods (*p* = 0.974). Similarly, for the state STAI post hoc analysis, there was a statistically significant increase in the incidence of anxiety in the pandemic group compared with the prepandemic (*p* < 0.001) and postvaccination (*p* < 0.001) groups, but no difference was found between the prepandemic and postvaccination groups (*p* = 0.512). Notably, the pandemic group was the only group with no outliers with elevated values (Figure 2). A post hoc comparison of those variables with a statistically significant main effect can be found in Appendix A. The corresponding figure of each statistical significance can be found in Appendix A.

## 4. Discussion

This prospective, simulation-based clinical cohort study was carried out with sixth-year undergraduate medical students during three different time periods (prepandemic, pandemic, and postvaccination) to compare anxiety levels before and after the randomized sham clinical simulation. The STAI revealed substantial dissimilarities among the pandemic clusters compared with the other two analyzed clusters concerning traits as well as state subscales, confirming the former assumption that the COVID-19 pandemic was a major disruption across multiple dimensions, including naturally, undergraduate medical education. These findings will help make simulation sessions more effective and identify external stressors that have not been assessed until now and that may influence students.

CS is a well-known teaching skill associated with a remarkable load of anxiety and unpredictability. Students must operate way outside of their comfort zone, be observed by classmates and facilitators, have unfamiliar conditions and equipment, and be generally in critical scenarios ([24]). Moreover, stressors such as these can contribute to a breeding ground that increases baseline anxiety levels and, in particular, to the results of simulations performed during the prevaccination phase of COVID-19, with the added guesswork generated by the pandemic response ([21]). Fredericks, S. et al. reported how high-fidelity scenarios of critically ill patients can induce emotional arousal in students and impact their cognitive load ([12]). Bommer, C. et al. revealed a significant anxiety burden linked to CS and revealed how prescenario briefing can alleviate anxiety, increase confidence, and improve performance ([6]). Barbadoro, P. et al. demonstrated that stress was increased more by simply being in a simulation scenario than by the intrinsic complexity of the required duty ([5]).

Therefore, CS, as a pedagogical technique, is well-established and educationally effective, with an intrinsic burden of associated anxiety from which we cannot disengage. CS provides a learning tool that uses simulators and anatomical models to recreate clinical scenarios with great accuracy ([33]), from critical situations that are difficult to reproduce to simpler scenarios where the evaluation of the situation is the fundamental skill to be developed. In this way, CS promotes quick decision-making, teamwork, self-learning, and self-criticism ([19]).

From a pedagogical point of view, CS represents a minor revolution in the traditional educational model. The student is going to have training experience in the simulation laboratory; it is not just a question of automatically performing techniques and procedures; rather, the student-instructor tandem must understand CS cases as reflective-experiential practices. The most reliable possible recreation of real situations (environment, furniture, high-performance simulator based on a physiological model, apparatus, real consumables, communications, complementary tests, etc.) makes it possible to experience these situations, which would not be possible otherwise, and in safe conditions, both for the patient and for the students ([20]).

In this sense, during the pandemic phase, in addition to anxiety related to CS, the uncertainty of COVID-19 evolution and the mandatory adoption of biosafety protocols to minimize risks were supplementary stressors ([27]). As demonstrated by health care providers, the use of personal protective equipment (PPE), such as FFP2 facemasks, gloves, protective suits, aprons, and face shields, is a standardized workflow procedure for achieving appropriate biosafety conditions to manage infectious patients ([22]) and is known to trigger an overactive physiological and stress-related response ([23]; [36]).

Anxiety levels (both as a state and as a trait) were significantly greater in the pandemic cohort. These disparities could not be attributed solely to isolated, individual, or interpersonal factors but rather to the simultaneous combination of time and space of a range of diverse factors with complex backgrounds, e.g., perceived fear of contracting disease, COVID-19 exposure during clinical fellowship training, limitations and/or social distancing, changes in education and curricular programs with the discontinuation of hospital practicums, information overload and worrisome news ([30]; [2]; [35]). Simulation experience, in turn, may smooth anxiety levels and improve performance in clinical situations ([6]; [32]). However, in our study, the influence of COVID-19 overrode previous experience. In the prepandemic cohort, no students reported previous simulation experience, in contrast to the 67.1% reported in the pandemic cohort and the 69.4% reported in the postvaccination cohort; however, the pandemic cohort scored significantly above anxiety rates, supporting the hypothesis concerning the relevance and disruption of COVID-19 in medical education. Similarly, no significant differences were found concerning the simulation case conducted (eight different scenarios), sex at birth, or role in the simulation (leader or assistant); thus, the independent significance associated with COVID-19 seems to be established ([18]).

An additional determinant of anxiety levels may be pandemic fatigue ([28]). The prepandemic cohort and postvaccination cohort ratings on the STAI questionnaire (both trait and state) were comparable. Notably, the incidence of outliers in the pandemic cohort was significantly lower than that in the remaining cohort. These data may suggest that undergraduate medical students maintain a uniform anxiety level during the first pandemic peak and support the assumption that herd immunity resulting from mass vaccination decreases anxiety levels by increasing safety ([31]). In fact, the results revealed similar anxiety values in the prepandemic and postvaccination periods, once again highlighting the crucial importance of COVID-19.

This study has several limitations. First, the data extractors were not blinded. To avoid cross-contamination, the associated partners were mandatorily trained face-to-face on handling the data collection notebook, randomization, and vital sign devices but were unaware that the STAI questionnaire was being used and that the score was derived. Only the principal investigator and data manager had full access to the complete database. Second, the STAI questionnaire was used as a tool for assessing anxiety levels, and the investigators were aware that there were other instruments or tests available for consideration. The STAI was chosen because it is a self-administered questionnaire validated in the Spanish version that has been widely used in a variety of contexts and has excellent internal consistency ([7]; [14]). In future studies, different methods for determining anxiety levels may be of great interest, e.g., the 7-item Generalized Anxiety Disorder Scale (GAD-7) ([37]) or the 9-item Patient Health Questionnaire (PHQ-9) ([15]). Third, vital signs (oxygen saturation; perfusion index; systolic, diastolic, and mean blood pressure; and heart rate and temperature) were measured as noninvasive, quick, and inexpensive methods to prevent unobtrusive interference during clinical simulation. Future studies could be very valuable for exploring the relationship between cortisol (or other biomarkers) and anxiety. In the present study, especially with respect to the pandemic cohort, invasive methods or techniques involving fluids (blood or saliva) to determine specific biomarkers were particularly inappropriate. Fourth, critical information such as the level of exposure to COVID-19 or the previous mental health of the participants was not collected; therefore, the lack of these confounding factors should be considered when interpreting the results. In this sense, comparisons to explore differences between subgroups, such as gender or previous simulation experience, that could influence anxiety levels were not performed since, for the first case, no previous statistically significant effects were found, and for the other variables, the differences were inherent to the nature of the study. Finally, the sample selection may be overly standardized. The participants were randomly recruited from among all the undergraduate medical students in their final year of medical school. The participants were students who had previous hands-on clinical experience in hospital residencies, primary care, or emergency medical services. Therefore, all the participants had fundamental knowledge about the initial management of critical patients and were familiar with the specific equipment and tools available in the laboratory simulation. This voluntarily imposed bias was designed to generate a homogeneous cohort to be assessed fairly accurately and where the educational background was comparable so that the influence on anxiety values would be consistent. In line with this last limitation, it is necessary to mention that the sample was obtained by a criterion of opportunity from a single university. However, in future studies, we expect to compare anxiety levels among different healthcare students (nurses, midwives, general practitioners, etc.) and different universities. Few studies have been conducted on the level of anxiety in undergraduate medical students who are exposed to the effects of COVID-19. Future studies analyzing this stressor or similar stressors that may arise and become a reality in students’ lives are essential for adequately planning simulation sessions.

## 5. Conclusions

In summary, the COVID-19 pandemic opened a novel and unprecedented paradigm, disrupting healthcare and health sciences education. Under these unique circumstances, medical students had to complete studies under very challenging biosafety conditions and with noticeable uncertainty about the progression of the pandemic. Anxiety levels in the pandemic cohort were substantially elevated, regardless of previous experience in simulation, the role adopted (leader or assistant), or the randomized scenario was undertaken. In addition, these results may contribute to understanding the phenomenon encountered during the hard phase of the COVID-19 pandemic and could serve as training for similar situations in the future. Performing CS in biohazardous environments significantly increases anxiety levels, so establishing mitigating measures to minimize the undesired effects of anxiety and promote the simulation-based learning process is necessary.

Understanding the factors that directly affect anxiety levels, and therefore the efficiency of the proposed simulation session can be of great interest to education managers and teachers, helping them design curricular practices on the basis of clinical simulation in a more proportionate way.

## Figures and Tables

**Figure 1 behavsci-15-00447-f001:**
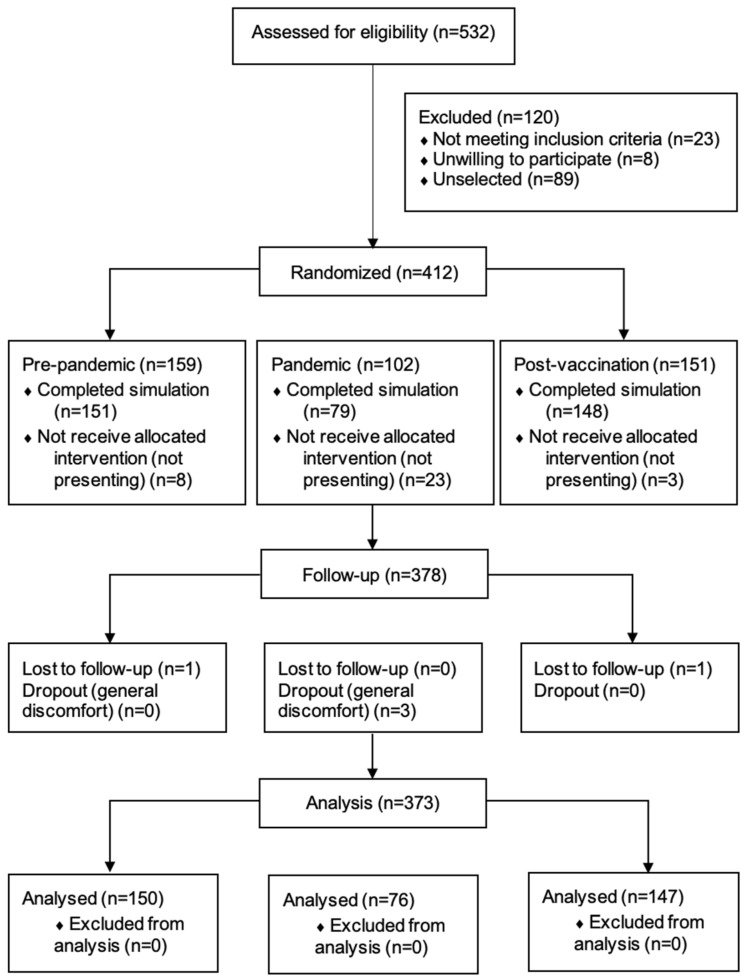
Diagram of study participation.

**Figure 2 behavsci-15-00447-f002:**
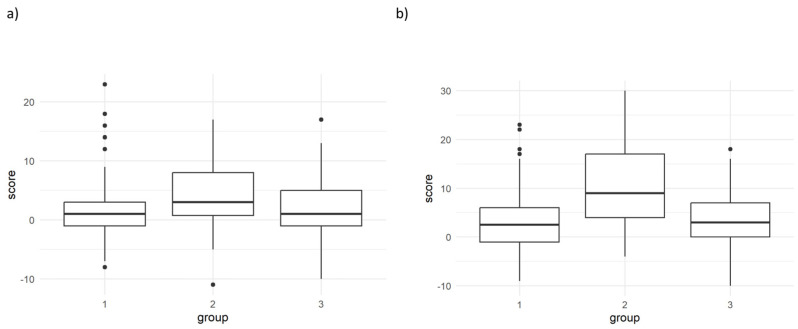
Differences pre- and posttraining for the trait STAI (**a**) and state STAI (**b**). Group 1 = prepandemic, 2 = pandemic, 3 = postvaccination. For the trait STAI, pandemic vs prepandemic (*p* < 0.001), postvaccination (*p* = 0.001), and prepandemic vs postvaccination (*p* = 0.974) periods were compared. For the state, the STAI pandemic was compared with the prepandemic (*p* < 0.001), postvaccination (*p* < 0.001), and prepandemic vs. postvaccination (*p* = 0.512) groups.

**Table 1 behavsci-15-00447-t001:** Descriptive data by study period.

	Study Period			
Variable ^a^	Prepandemic	Pandemic	Postvaccination	*p* Value ^b^
No. (%) with data	150 (40.2)	76 (20.4)	147 (39.4)	N.A.
Sex at birth, female	99 (66)	54 (71.1)	103 (70.1)	0.660
Age, year	23 (23–24)	24 (23–24)	23 (23–24)	0.338
Body mass index, kg/m^2^	21.6 (19.8–23.3)	22.2 (20.1–24.7)	21.1 (19.5–23.2)	0.078
Lifestyle habits				
Smoking	20 (13.3)	15 (19.7)	7 (4.76)	0.002
Coffee/tea	130 (86.7)	64 (84.2)	110 (74.8)	0.025
Energy drinks	9 (6)	12 (15.8)	8 (5.44)	0.014
Prior simulation experience, yes	0 (0)	51 (67.1)	102 (69.4)	<0.001
Leader, yes	78 (52)	41 (53.9)	78 (53.1)	0.960
Randomization				0.998
Hip fracture	20 (13.3)	11 (14.5)	20 (13.6)	
Asthmatic exacerbation	20 (13.3)	10 (13.2)	20 (13.6)	
Sepsis	20 (13.3)	9 (11.8)	20 (13.6)	
Acute heart failure	18 (12.0)	7 (9.21)	17 (11.6)	
Polytraumatized	20 (13.3)	11 (14.5)	18 (12.2)	
Anaphylactic shock	20 (13.3)	11 (14.5)	18 (12.2)	
Acute myocardial infarction	19 (12.7)	10 (13.2)	19 (12.9)	
Thromboembolism in PW	13 (8.67)	7 (9.21)	15 (10.2)	
Baseline evaluation				
Vital signs				
Oxygen saturation, %	98 (97–98)	98 (97–98)	98 (97–98)	0.743
Perfusion index, %	2.1 (1.1–4.4)	2.1 (1–4.3)	1.9 (0.9–4.4)	0.989
SBP, mmHg	130 (122–146)	125 (115–136)	134 (124–154)	<0.001
DBP, mmHg	84 (75–91)	76 (71–86)	84 (77–90)	0.006
MBP, mmHg	100 (92–110)	92 (87–102)	102 (93–112)	<0.001
Heart rate, beats/min	84 (74–96)	91 (74–104)	93 (80–111)	0.002
Temperature, °C	37 (36.8–37.3)	37.1 (36.6–37.3)	36.9 (36.6–37.2)	0.076
Postsimulation evaluation				
Vital signs				
Oxygen saturation, %	98 (97–99)	98 (97–99)	98 (97–99)	0.905
Perfusion index, %	1.4 (0.9–2.4)	1.3 (0.7–2.3)	1.4 (0.8–2.4)	0.787
SBP, mmHg	130 (121–149)	132 (122–137)	133 (123–160)	0.005
DBP, mmHg	83 (76–91)	84 (79–91)	83 (77–92)	0.932
MBP, mmHg	101 (91–113)	101 (95–105)	104 (90–114)	0.206
Heart rate, beats/min	85 (74–98)	88 (75–100)	91 (80–114)	0.001
Temperature, °C	36.9 (36.7–37.2)	36.9 (36.6–37.2)	36.8 (36.6–37.1)	0.044

Abbreviations: NA: not applicable; SBP: systolic blood pressure; DBP: diastolic blood pressure; MBP: mean blood pressure; PW: pregnant woman. ^a^ Values are expressed as the total number (percentage) and median (25th–75th percentile), as appropriate. ^b^ The Mann–Whitney U test or chi-square test was used as appropriate.

**Table 2 behavsci-15-00447-t002:** Anxiety level by the group during the study period.

	Study Period			
Variable ^a^	Prepandemic	Pandemic	Postvaccination	*p* Value ^b^
No. (%) with data	150 (40.2)	76 (20.4)	147 (39.4)	N.A.
Baseline evaluation				
State-Trait Anxiety Inventory				
State, points	41 (37–47)	57 (50–63)	41 (37–45)	<0.001
Trait, points	31 (27–37)	41 (33–48)	32 (29–37)	<0.001
Postsimulation evaluation				
State-Trait Anxiety Inventory				
State, points	45 (40–50)	66 (62–71)	46 (41–50)	<0.001
Trait, points	32 (27–39)	47 (40–51)	34 (29–39)	<0.001
State-Trait Anxiety Inventory difference				
State, points	2.5 (−1 to 6)	9 (4–17)	3 (0–7)	<0.001
Trait, points	1 (−1 to 3)	3 (0.2–8)	1 (−1 to 5)	<0.001

Abbreviations: NA: not applicable. ^a^ Values are expressed as the total number (percentage) and median (25th–75th percentile), as appropriate. ^b^ The Mann–Whitney U test or chi-square test was used as appropriate.

**Table 3 behavsci-15-00447-t003:** Multinomial logistic regression.

Variable ^a^	Estimate	Standard Error	Z Value	*p* Value
Previous experience: prepandemic	20.5	831.8	NA	NA
Previous experience: postvaccination	−0.23	0.36	−0.66	0.508
SBP: prepandemic	0.04	0.02	2.04	0.040
SBP: postvaccination	0.05	0.01	NA	NA
MBP: prepandemic	−0.02	0.02	−0.81	0.415
MBP: postvaccination	−0.01	0.02	−0.72	0.466
STAI-State: prepandemic	−0.17	0.03	−5.43	<0.001
STAI-State: postvaccination	−0.12	0.02	−4.63	<0.001
STAI-Trait: prepandemic	−0.001	0.04	−0.03	0.974
STAI-Trait: postvaccination	−0.03	0.03	−1.03	0.302

Abbreviations: SBP: systolic blood pressure; MBP: mean blood pressure; STAI-State: differences pre/post for State-STAI; STAI-Trait: differences pre/post for trait-STAI. ^a^ The group name following the variable name refers to the time period for which the comparison was performed. The reference group is the pandemic time period.

## Data Availability

The data presented in this study are available upon request from the corresponding author. The data are not publicly available due to privacy concerns.

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
