# Peer review of "A Comparison of Pre- and Post-Clinical Simulation Anxiety Levels of Undergraduate Medical Students Before and During the COVID-19 Pandemic: A Prospective Cohort Study"

_behavsci, 2025, doi:10.3390/bs15040447_

Round 1
Reviewer 1 Report
Comments and Suggestions for Authors
REVIEW REPORT (behavsci-3502958)
Face-to-face comparison of anxiety levels during high-fidelity clinical simulations of undergraduate medical students before, during, and after the COVID-19 pandemic
Dear Authors,
Congratulations on completing this manuscript. Please understand that the comments below are exclusively related to the final improvement of the document. Please highlight the changes in the version to be submitted in a different color and please do not use Word's change tracking feature to better identify the changes.
Best regards
--------------------------------------
ABSTRACT
- The abstract provides a good, succinct reflection of the study as a whole.
INTRODUCTION
- The authors provide a good contextualization of the impact of COVID-19 on medical education, and this is a positive point. But why exactly is student anxiety during clinical simulation a relevant problem? The authors need to make this very clear.
- Anxiety is an important and very current problem in times of excessive consumption of technology (including in the academic environment). Convinced that this study has great potential, I ask the authors to clearly define the constructs of state anxiety and trait anxiety, which are central to the study.
- Still related to the previous point (anxiety in clinical simulations), there is a lack of previous studies that address this problem in students during the pandemic, showing how anxiety has been studied in this context previously. The authors should review the introduction to make it clear that this problem is important. - After reading the introduction more than once, I was left with the following question: what aspects of this study are novel and how do they advance current knowledge on the topic?
METHODS
- The research was conducted with medical students from a single university. If this was not included in the study limitations, please include it. Also, objectively justify why only students from the University of Valladolid were included.
- Do the authors have information on confounding variables, such as previous experience in simulations, level of exposure to COVID-19, or previous mental health of the participants? If this information is not available, include it in the study limitations at the end of the discussion.
- If there was randomization of the scenarios, as indicated in the text, the authors should detail how the comparison in terms of baseline characteristics was ensured. In addition, the authors should include more information on the content and duration of the training of the evaluators.
RESULTS
- Is it possible to conduct analyses to explore differences between subgroups, such as gender or previous simulation experience, that could influence anxiety levels?
DISCUSSION
- In the first paragraph, in addition to highlighting the main findings, also include two or three sentences that reflect the main practical implications of the results.
- I have the impression that it was difficult to compare the results with previous studies, probably due to the limited literature. Therefore, emphasize the importance of future research.
- Can the differences between the groups be explained, in part, by the lack of uncollected information (e.g., previous anxiety, use of anxiolytics, etc.)?
- In the limitations, in addition to those mentioned above, please include the lack of diversity in the sample or uncontrolled contextual factors that may have affected the results.
CONCLUSIONS
- How do the authors suggest the results be applied in practice, especially in crisis contexts?
Author Response
We would like to thank you for the extensive review of our manuscript. We hope that our responses below and the changes derived from them improved the manuscript. The answers are in the attached word document.

Reviewer 2 Report
Comments and Suggestions for Authors
Dear Authors,
Thank you for allowing me to review your manuscript. Your study is interesting and an important topic to be considered when designing simulations for health professions students.
General considerations
Strengths
- The study addresses a relevant and timely topic in clinical simulation
- The quantitative methodology is appropriate for assessing the anxiety levels of students
- The manuscript presents a clear and logical structure.
- Results are presented in a detailed and well-substantiated manner.
Weaknesses:
- Some sections could benefit from more in-depth discussion.
Suggestions for Improvement:
- Authors need to be more consistent with terminology. Different terms such as simulation, intervention, and training have been used throughout the paper which makes it confusing.
By Sections:
- Title:
Strengths:
- The title clearly indicates the main focus of the study: comparing anxiety levels in undergraduate medical students
- It specifies that Covid-19 pandemic may have had an influence on the results.
Weaknesses:
- The title does not clearly state the design of the study
- The title does not clearly indicate what comparisons are taking place.
Suggestions for Improvement:
An alternative to the title could be:
“A Prospective Cohort Study on Anxiety Levels in Medical Students During Clinical Simulations: Prepandemic, Pandemic, and Postvaccination Trends”
OR
“A comparison of pre-and post-clinical simulation anxiety levels of undergraduate medical students before and during Covid-19 pandemic: A prospective cohort study”
- Abstract:
Strengths:
- The abstract provides a clear and concise aim of the study and the design of the study
- It clearly states the purpose of the study and its relevance in the context of the COVID-19 pandemic.
- The primary outcome of the study is clearly stated in the abstract
Weakness/Suggestions for Improvement:
- Line 33-34: The authors state “Patient demographics and baseline vital signs (before CS) were collected”. It is not clear with this statement whether the study was conducted on students or patients. I recommend that authors use consistent terminology.
- Lines 35-38: It is not clear what the authors are stating here. Is it “There is a statistically significant increase in the incidence of anxiety during pandemic time period when compared to prepandemic and post vaccination?”
- Consider adding a brief statement about the limitations of the study.
- If possible, include a sentence about the specific types of simulations used in the study.
- Line 40 – conclusion/implication: Should be clear. You can use the same conclusion that you had at the end of your manuscript.
Introduction
Strengths
- Adequate contextualization of the influence of Covid-19 on education.
- Clear presentation of the study
Weaknesses/Suggestions for Improvement:
- Include more literature on anxiety during clinical simulation
- Strengthen the theoretical basis with more references to educational theories underpinning simulations.
- Include information regarding anxiety in students related to clinical simulations
- Materials and Methods
Strengths
- Study design is described well.
- Study conforms to STROBE guidelines
- Biosafety protocol described well
- Study protocol and statistical analysis sections are explained well
Weakness/Suggestions for improvement
- Line 93-105: While the authors mentioned the biosafety protocol during the pandemic, it would be helpful to mention what the normal prepandemic or postvaccination protocol is followed in the clinical sim lab.
- Participants and randomization: How were the students recruited? Is this simulation a mandatory activity? Please clarify. Were there any incentives?
- Sample size calculation is in the supplementary file, but a few sentences on how the sample size was determined will be helpful for the readers.
- It is not clear why students completing the STAI during the last year were excluded (Line 110). Also not clear are the details of whether the students were asked about baseline history of preexisting anxiety, which could skew the results.
- Study protocol (section 2.5): Was the survey given electronically or paper? This should be specified. How was the data secured? What was the process of deidentification?
- Line 141: Was the direct feedback given by the facilitators or to the facilitators?
- The authors should include more information regarding the simulation process to clarify the fidelity and the intensity of the simulation and its effect on anxiety scores of the students.
- Ethical Aspects:
Strengths:
- Clear mention of IRB approval with protocol number.
- Explicit statement about obtaining informed consent.
- Mention of written informed consent.
Suggestions for Improvement:
- Include a brief description of the informed consent obtaining process.
- Clarify ethical considerations for groups with mandatory participation (if applicable in this study)
- Results:
Strengths:
- Clear and logical organization of results.
- Effective use of quotations to illustrate themes.
- Objective presentation, with quantitative support where appropriate.
- Use of tables and figures is very good for data visualization
Suggestions for Improvement:
- Figure 1 – “Received allocated intervention” needs to be reworded. The simulation would not be considered an intervention. Maybe you can reword to “participated in simulation or completed simulation”.
- Figure 1: What does “not presenting” mean? Were the students absent?
- Primary outcome – Line 183. “The STAI trial results were smoother..” could be changed or defined better to explain what smoother means.
- Table 2: What does previous experience mean? Students with prior experience in simulation?
- Figure 2 – explanation – Lines 207-214: There is a mention of increased incidence of COVID-19 between the periods. This is confusing. Was the incidence of COVID-19 being studied? If so, this was not mentioned in the methodology section.
- Discussion
Strengths:
- Comprehensive discussion addressing all main findings of the study.
- Good integration with existing literature.
- Consideration of practical implications of the results.
- Good discussion and clear acknowledgement of the limitation
Suggestions for Improvement:
- Line 233-236: This information regarding anxiety related to clinical simulations should also be in the introduction paragraph
- Expand the discussion on theoretical implications of findings. Clinical simulation as a pedagogical technique is mentioned in lines 247-248 but could be elaborated.
- Lines 278 – 281: The authors assume that the herd immunity from mass vaccination may be the reason for decreased anxiety levels. Instead of stating the “data suggest”, you may want to say “data may suggest” since no explicit questions were asked of the students regarding this.
- The authors did not mention how they controlled for the biases mentioned.
- Hypothesis was mentioned in the discussion (269-270), but not seen in the methodology section
- Conclusions:
- Strengths:
- Clear and well-articulated conclusions.
- Strong connection between conclusions and study results.
- Consideration of practical implications of findings.
- Suggestions for Improvement:
- Briefly expand on how results contribute to theoretical advancement in the field of clinical simulation
- Consider adding a brief statement on the generalizability of results to other educational contexts.
- Include more specific suggestions for future research.
Comments on the Quality of English Language
The English could be improved to more clearly express the research.
Author Response

(The authors gave the same response as above.)

Round 2
Reviewer 1 Report
Comments and Suggestions for Authors
None
Reviewer 2 Report
Comments and Suggestions for Authors
Thank you for responding to the review and submitting revisions.